# LUMA: A Benchmark Dataset for Learning from Uncertain and Multimodal Data

## Abstract

Multimodal Deep Learning enhances decision-making by integrating diverse information sources, such as texts, images, audio, and videos. To develop trustworthy multimodal approaches, it is essential to understand how uncertainty impacts these models. We propose LUMA, a unique benchmark dataset, featuring audio, image, and textual data from 50 classes, for learning from uncertain and multimodal data. It extends the well-known CIFAR 10/100 dataset with audio samples extracted from three audio corpora, and text data generated using the Gemma-7B Large Language Model (LLM). The LUMA dataset enables the controlled injection of varying types and degrees of uncertainty to achieve and tailor specific experiments and benchmarking initiatives. LUMA is also available as a Python package including the functions for generating multiple variants of the dataset with controlling the diversity of the data, the amount of noise for each modality, and adding out-of-distribution samples. A baseline pre-trained model is also provided alongside three uncertainty quantification methods: Monte-Carlo Dropout, Deep Ensemble, and Reliable Conflictive Multi-View Learning. This comprehensive dataset and its benchmarking tools are intended to promote and support the development, evaluation, and benchmarking of trustworthy and robust multimodal deep learning approaches. We anticipate that the LUMA dataset will help the ICLR community to design more trustworthy and robust machine learning approaches for safety critical applications.

## 1 Introduction

In recent years, the use of Machine Learning and Deep Learning has surged across various fields, driving advancements in data analysis and decision-making. In domains such as healthcare, autonomous driving, and finance, information is distributed across multiple modalities including audio, video, text, and images. To better understand the data and improve decision-making capabilities, it is crucial for deep learning models to integrate diverse, multimodal sources of information. Multimodal Deep Learning (MDL) addresses this need and improves the capabilities of uni-modal networks (Bayoudh et al., 2022; Krones et al., 2024; Xiao et al., 2020; Lee & Yoo, 2020).

Another important consideration for deploying deep learning models in safety critical fields is trustworthiness. Traditional deep learning models are often overconfident in their predictions (Abdar et al., 2021), which can lead to catastrophic results in areas such as healthcare or autonomous driving. Although various techniques for uncertainty quantification have been proposed to measure the level of uncertainty in data and model, this remains an open and challenging area. More research and robust benchmarks are needed to advance the field of uncertainty quantification in deep learning (Krishnan & Tickoo, 2020; Nado et al., 2021).

In probabilistic modeling, uncertainty is usually divided into aleatoric (data) and epistemic (model) uncertainties (Kiureghian & Ditlevsen, 2009). Aleatoric uncertainty refers to the uncertainty in the data due to inherent noise. It is impossible to reduce the amount of aleatoric uncertainty with additional data (hence, it is also often called irreducible uncertainty). Epistemic uncertainty is the uncertainty in model parameters, due to lack of data, hence, it can be reduced with additional data samples. Epistemic uncertainty is also usually high for Out-of-distribution (OOD) data, and is commonly used for OOD detection.

Multimodal uncertainty quantification (MUQ) is a relatively new research area that adapts uncertainty quantification approaches to multimodal deep learning problems, aiming to enhance the trustworthiness of these models (Jung et al., 2022). Due to the unsupervised nature of uncertainty quantification, where the exact extent of uncertainty in the data and the model is unknown, analyzing and benchmarking proposed UQ methods is challenging. Current multimodal datasets used for benchmarking state-of-the-art models in multimodal uncertainty quantification (Han et al., 2023; Jung et al., 2022; 2023; Liu et al., 2022; Xu et al., 2024) lack the ability to inject a controlled amount and various types of uncertainties for each modality. This limitation hinders the comprehensive benchmarking of MUQ techniques, which is essential for developing trustworthy and robust multimodal deep learning approaches.

To address this challenge, we introduce LUMA (**L**earning from **U**ncertain and **M**ultimodal d**A**ta), a multimodal dataset specifically designed for benchmarking multimodal learning algorithms on uncertain data. The dataset includes 101,000 images, 135,096 audio recordings, and 62,875 text passages, amounting to approximately 3 GB of data. Each modality is independently sourced, reflecting real-world conditions where data is often collected under different conditions and times. For example, in medical contexts, diagnostic data from different modalities such as radiography, MRI, and ECG/EEG are gathered asynchronously, leading to modality-specific uncertainties. The modalities are carefully aligned, ensuring that each text passage is related to the object in the corresponding image, and each audio recording is the pronunciation of the object label in the image. The provided Python toolkit allows the injection of aleatoric and epistemic uncertainties in a controlled and parameterized way into each modality specifically.

To summarize, our contributions are as follows:

1. We propose LUMA[1], a benchmarking dataset for learning from uncertain and multimodal data. It includes audio, image, and textual modalities across 50 distinct classes. We compiled the images from the CIFAR 10/100 dataset (Krizhevsky, 2009), extracted, validated, and associated the corresponding audio samples from three diverse audio corpora, and generated the related text modality utilizing Gemma-7B Instruct (Mesnard et al., 2024) Large Language Model (LLM). We also performed additional bias analysis of the dataset. Each generated version of the dataset consists of 600 data records per class (500 for training, and 100 for testing) belonging to 42 classes, and 3,859 OOD data points, belonging to the remaining 8 classes.

2. We offer a Python package [1] that generates dataset samples with varying levels of noise and uncertainty. The uncertainty generator can effectively increase aleatoric uncertainty in the data and epistemic uncertainty in the model.

3. Finally, we provide baseline models including three different uncertainly quantification methods (Monte-Carlo Dropout (Gal & Ghahramani, 2016), Deep Ensemble (Lakshminarayanan et al., 2017), Reliable Conflictive Multi-View Learning (Xu et al., 2024)), to serve as a starting point for benchmarking.

## 2 LIMITATIONS OF CURRENT DATASETS FOR MDL BENCHMARKING

In practice, we often don't know the extent of inherent uncertainties in the data or how accurately they represent the real-world data space. This often makes it hard to evaluate how well uncertainty quantification algorithms work. Moreover, deep learning algorithms may behave differently under different amount of uncertainties (i.e., the robustness to noise may vary). Thus, it may be beneficial to inject additional amount of noise in the data, and observe the change in uncertainty metrics and the performance of the models. Since approaches to quantify different types of uncertainty vary, it is beneficial to have options for injecting various types of uncertainties.

Several datasets are used in multimodal uncertainty quantification settings. A notable line of work (Han et al., 2023; Jung et al., 2022; 2023) has employed datasets such as HandWritten[2], CUB[3],

---

[1]https://osf.io/8ph6y/?view_only=8272969d8cd34c0b9459659fb4f41507 (Anonymized for the peer-review process. Will be substituted with public links in case of acceptance.)

[2]https://archive.ics.uci.edu/ml/datasets/Multiple+Features

[3]http://www.vision.caltech.edu/visipedia/CUB-200.html

Scene15[4], and Caltech101 [5]. These datasets typically extract different features from unimodal sources to create a multi-view setup. While they have been instrumental, they primarily repurpose unimodal data for multimodal tasks, underscoring the need for more comprehensive and inherently multimodal datasets to better evaluate uncertainty in deep learning models.

Furthermore, the current approaches that introduce uncertainty in the data (Han et al., 2023; Jung et al., 2022; 2023) add Gaussian noise to the views or the extracted features. While Gaussian noise does increase uncertainty, it does not accurately reflect the noise that can be found in real-world datasets and this process lacks fine-grained control over the type of uncertainty being injected.

Additionally, how different modalities' uncertainties interact significantly impacts the overall multimodal uncertainty. When both modalities encode redundant information, the total uncertainty might not decrease. Conversely, conflicting information can lead to increased uncertainty, while complementary information can reduce it. A deeper understanding of these phenomena is crucial. Fine-grained control over individual modalities' uncertainties opens the way for more theoretical research based on empirical observations.

To better understand and analyze uncertain multimodal data, as well as to debug and benchmark uncertainty quantification techniques in the multimodal learning context, we propose a dataset accompanied by an uncertainty generator package. This package includes various techniques for injecting uncertainty, such as controlling data diversity, adding different types of real-world noise, randomly switching labels to their closest class, and injecting out-of-distribution (OOD) data.

## 3 LUMA DATASET

In this section, we introduce LUMA, a dataset composed of an extensible list of modalities including image, audio, and text modalities, collected from various sources.

### 3.1 IMAGE MODALITY

For the image modality, our priority was to choose a relatively simple yet well-known dataset, where we could have the option to manually increase the degree of uncertainty. For that purpose, we chose CIFAR-100 and CIFAR-10 (Krizhevsky, 2009) datasets since they are well-known datasets of small 32x32 images, with lots of baseline models. 42 classes were chosen so that after aligning with the other modalities, we would have at least 600 samples in each class per modality. The 600 threshold was selected based on the number of images per class in the CIFAR-100 dataset. We took another 8 classes, which had less than 600 samples after aligning with other modalities, as OOD samples. In total, we took 25,200 images as train/test data, and 3,859 images as OOD data (see the image collection pipeline in Figure 1).

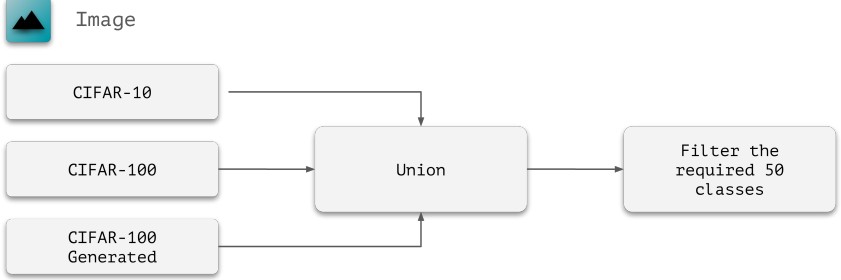

Figure 1: Image collection pipeline

Aside from the main dataset, as described in Section 3.4, another priority was to understand the behaviors of models under different levels of data diversity. To achieve this, we decided to sample 600 data points with different level of diversity from the bigger set CIFAR-10/100. However, in CIFAR-100 dataset, there are no more than 600 samples per class. We alleviated this issue with

---

[4]https://serre-lab.clps.brown.edu/resource/hmdb-a-largehuman-motion-database
[5]https://data.caltech.edu/records/mzrjq-6wc02

including images generated with EDM Diffusion-based generative model[6](Karras et al., 2022). We chose EDM-generated images, since the generated samples were already available, and Zheng et al. (2024) showed that augmenting CIFAR-10 data with EDM-generated samples significantly improves the classification accuracy.

## 3.2 AUDIO MODALITY

For audio modality, the diversity of accent in the pronunciation was an important factor to be considered and we collected samples, where different people would pronounce the corresponding class label. For this task, we used three audio/text parallel corpora, and extracted the desired audio segments. More specifically, we used The Spoken Wikipedia (Köhn et al., 2016), LibriSpeech (Panayotov et al., 2015), and Mozilla Common Voice (Ardila et al., 2020) corpora. The audio collection pipeline is depicted in Figure 2.

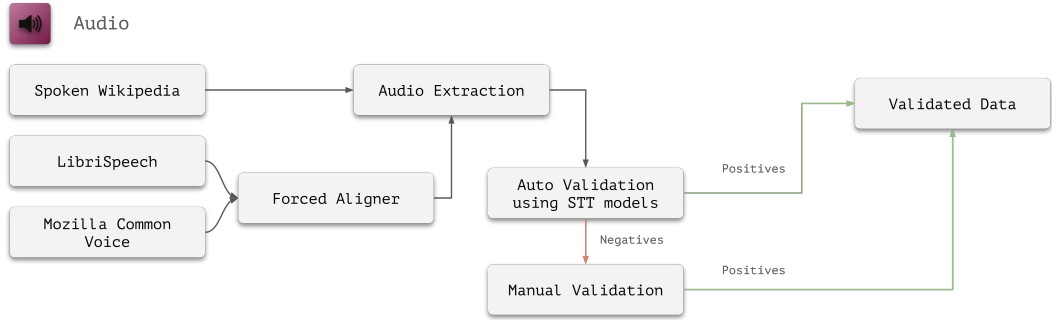

Figure 2: Audio collection, extraction and validation pipeline

The Spoken Wikipedia is a collection of hundreds hours of phoneme-level aligned audio, where volunteer readers are reading various Wikipedia articles. We used these alignments to extract all the instances of audio segments that pronounced one of the CIFAR-10/100 classes.

The LibriSpeech dataset is a corpus of 1,000 hours of English speech, derived from audiobooks from the LibriVox[7] project, which is a collection of public domain audiobooks. Unfortunately, LibriSpeech doesn't provide word-level alignment, hence, we used force-aligned alignments[8] generated with the Montreal Forced Aligner (McAuliffe et al., 2017). Similarly to The Spoken Wikipedia, we looked up the CIFAR-10/100 labels in forced aligned textual data, and extracted the corresponding audio segments.

The Mozilla Common Voice corpora, is a crowdsourced open-source collection of voices by volunteer contributors from around the world. Like LibriSpeech, Mozilla Common Voice also doesn't provide word-level alignments, hence, we again used forced aligned alignments[9], and extracted the relevant audio samples.

73 additional recordings of pronunciations belonging to 4 classes ("roses", "telephone", "whale", "wolf") were voluntarily contributed by our colleagues, which were anonymized, trimmed, and added to the dataset.

From these corpora, we used the following rule to extract the samples. First, we extended our class label set with a superset that also contains the plural forms of the words (i.e., for the audio track "horse", the audio track "horses" was added to the set), then we iterated over all aligned transcripts, and for any word included in the formed set, we extracted the corresponding audio sample. We considered the plural forms, since we believe that an extra "s" or "es" does not change the pronunciation of the words much. We did not consider the plural forms if it requires audible changes to the word root (i.e., mouse - mice). The extraction algorithm can be found in Appendix B.2.2.

---

[6]Retrieved generated samples from https://github.com/wzekai99/DM-Improves-AT

[7]https://librivox.org/

[8]https://github.com/CorentinJ/librispeech-alignments

[9]https://github.com/JRMeyer/common-voice-forced-alignments

Since most of the audio data is collected from forced alignments, it is possible to have misaligned audio segments, which could introduce additional noise to the dataset. Moreover, since part of the audio samples are from voluntary contributions, there can be very noisy samples, which are hard to interpret, or audio samples with a strong accent, which again can be hard to interpret. To remove such extreme cases in audio samples, we performed an automatic validation of the samples. Then, we filtered out the false negatives with manual validation for the negative predictions.

The automatic validation was achieved with the OpenAI's Whisper Large V3 model (Radford et al., 2023) for audio transcription, and transcribed the extracted audio samples. If the transcription corresponded to the class label (or its plural form), then we considered the sample as valid. Otherwise, the sample was sent for manual validation. Because of the huge output space of the Whisper Large V3 model, the probability of false positives is quite low, so we did not perform a manual validation for positive predictions. To summarize, we validated 130,069 out of 178,123 data samples with automatic validation, and we performed a manual validation for the remaining samples.

For manual validation, we decided to check only the classes, which did not have more than 800 samples (to be able to sample 600 samples with different degrees of diversity, as described in Section 3.4). Hence, we filtered 8,372 samples, and scheduled them for manual labeling. We opted for Label Studio (Tkachenko et al., 2020-2022) to build the labeling interface (see Appendix B.2.2 for the annotation interface). The interface provided the audio sample, with the prompt "Is the audio saying the word below? (An extra 's' or 'es' in the pronunciation is okay.)" and answer options of "Yes" or "No". We asked our colleagues (M.Sc. and PhD Students, and Professors) with advanced to fluent English knowledge to annotate the samples.

In total, we collected 2 annotations per sample, from 17 annotators. We got 71.61% of annotation agreement, and accepted 5,027 samples, where both annotators confirmed the validity of the sample. Hence, we took the 42 classes that had more than 600 validated samples (automatically and manually) as training/test data, and we took the remaining 8 classes as OOD data. In total, the auto-validated and manually validated audio samples combined, LUMA has 135,096 audio samples. The final distribution of audio data across classes can be seen in the Appendix B.2.2.

## 3.3 TEXT MODALITY

For text modality, the main constraint was that the text segments had to talk about the subject of the images. For that, we decided to employ a generative model, and generate text segments about the class label. We utilized Google's Gemma-7B Instruct model (Mesnard et al., 2024) to generate more than 1,200 texts samples per class, using 13 different prompts. Here is an example of a prompts used for generation (please find the full list of prompts in Appendix B.3).

```
"You are explaining a five year old child what the word <word>
    means. Use very simple and explanatory language, so the kid
    will understand the meaning of the word <word>. Tone: Casual,
    complexity: simple"
```

Gemma-7B Instruct was chosen, since according to their technical report (Mesnard et al., 2024), it outperforms other open LLMs with similar size, in 11 out of 18 tasks. Moreover, in our experiments, it provided better answers to our prompts compared with Mistral-7B (Jiang et al., 2023).

To validate that the generated texts accurately represent the labels, we masked all label occurrences in the text and fed the masked text back into the Gemma-7B Instruct model, asking it to classify the text into one of the labels (see the prompt in the Appendix B.3). Based on the prediction of the model, if the prediction matched the ground truth label, we accepted the sample as validated. In total, we accepted 55,953 text samples.

After manually analyzing some of the generated texts, we noticed that there were samples with offensive biases and stereotypes (some examples are included in the Appendix B.3). Particularly, we noticed a lots of gender bias for classes "man", "woman", "boy" and "girl". To find the proportion of the biased data, we asked the Gemma model to find out if the given text contains gender, racial, religious, or cultural biases. We found out, that indeed, the aforementioned 4 classes have a huge amount of gender bias (see the bias detection statistics in Appendix B.3). Our hypothesis is, that describing a man or woman in an unbiased way is a challenging task for LLM models (as well as for

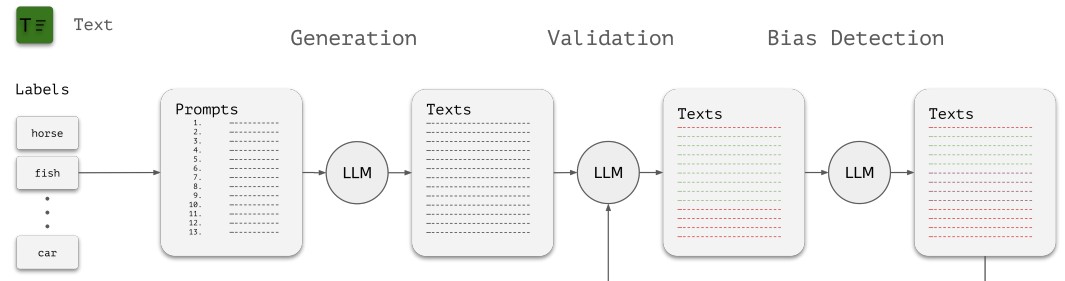

Figure 3: Text generation and validation pipeline

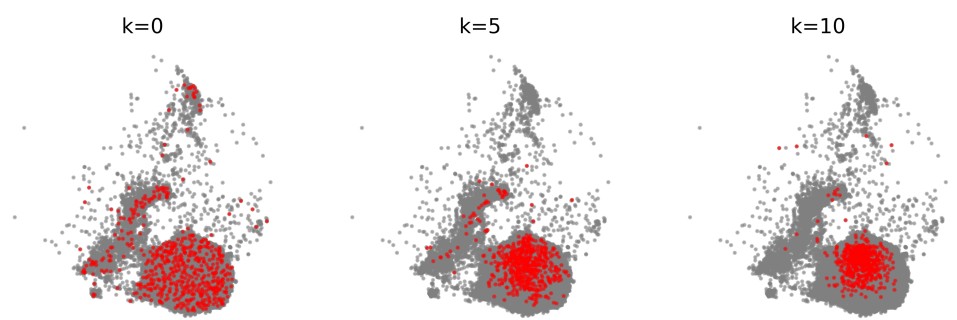

Figure 4: t-SNE (van der Maaten & Hinton, 2008) visualization of audio data points from class "man" (in gray), and sampled points with different diversity parameter $k$. The higher is the value of $k$, the more concentrated (less diverse) the points are.

humans), which are trained on unbalanced data (Kotek et al., 2023). For that purpose, we reconstruct the prompts (see in Appendix B.3), to explicitly provide topics and keywords with occupations, which will minimize the bias. We then rerun the bias detection prompt, which found fewer biased samples and allowed us to filter enough samples to be included in the final dataset. Since textual data was generated using an LLM, we recognize that the dataset may contain factual inaccuracies, or biases, but our aim is to offer a benchmark to study uncertainty quantification in multimodal classification settings. LUMA shall not be used as a source of knowledge or information.

### 3.4 DATASET COMPILATION

Based on the collected samples from the 3 modalities (image, text, audio), we wanted to compile a dataset with little uncertainties, and later, provide tool to inject uncertainties on demand. Our priority was to propose several options for uncertainty control and parameter setting: data diversity, sample noise, label noise, OOD injection.

**Data Diversity:** With a fixed number of data points, increasing the diversity of the data enhances the information passed to the model, thereby it shall reduce the epistemic uncertainty. Conversely, when samples are concentrated at a single point in the latent space, they encode less information, which shall lead to greater epistemic uncertainty in areas where data is scarce (Figure 5). Hence, controlling the diversity of the data allows us to study the behavior of epistemic uncertainties under varying amounts of information.

To control the diversity, we extract deep features from each modality (Wav2Vec (Baevski et al., 2020) for audio, BERT (Devlin et al., 2019b) for text and VGG-11 (Simonyan & Zisserman, 2015) for images (see Figure 6 for t-SNE visualizations of said features), and compute the inverse distance of each sample to the center (mean vector) of its class, raised to the power of $k$:

$$D_i = \frac{1}{\left\| F_i - \frac{1}{|C|} \sum_{j \in C} F_j \right\|_2^k}, \quad i \in C, \tag{1}$$

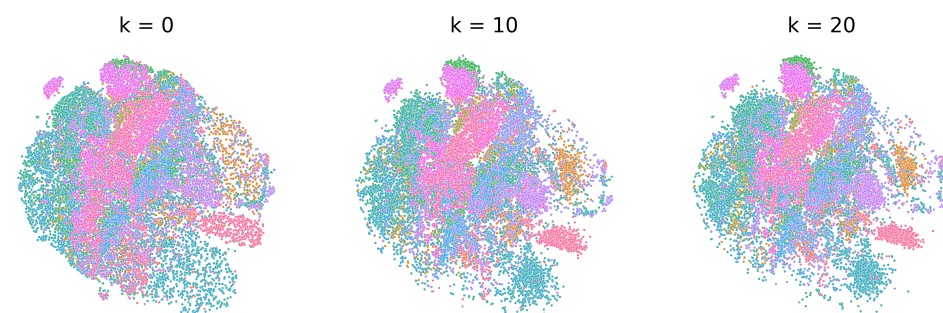

Figure 5: t-SNE (van der Maaten & Hinton, 2008) visualization of audio data points for all classes, sampled with different diversity parameter $k$. With higher $k$ we have more concentrated samples, and more separation between classes. The diversity can similarly be controlled for the other modalities.

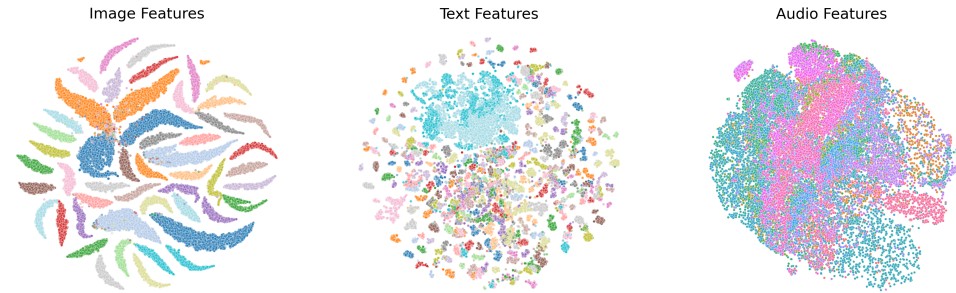

Figure 6: t-SNE (van der Maaten & Hinton, 2008) visualization of image, text and audio data points for all classes.

where $F$ represents the deep feature vectors extracted from the samples, $C$ is the set of data indices belonging to the class, and $|\cdot|$ measures the cardinality of the set. Then, having the inverse distances, we sample points from categorical distribution $x_n \sim Categorical(D)$. In Eq. 1, $k$ is the variable controlling the diversity. If $k = 0$ the sampling is uniform. The bigger $k$, the higher probability of selection will be applied to the samples closer to the center.

Having sufficient samples in image and text modalities, our bottleneck was the number of samples in the audio modality. Since in the 42 in-distribution classes, around 70% have more than 900 audio samples, we considered this enough for diversity control.

**Sample Noise**: We want to have an option to inject controlled amount of noise in the data. This may reduce the information in each data sample, and increase the classification difficulty. With our hypothesis, this may affect both epistemic and aleatoric uncertainty degrees. This type of noise can also be very beneficial for estimating the model robustness to noise. We apply different types of noise to each modality.

For **audio modality**, we added background noise from ESC-50 dataset (Piczak, 2015) to each sample, using the `audiomentations`[10] library. The amount of the minimum and maximum signal-to-noise ratio, as well as the proportion of the noisy data is set as a hyper-parameter.

For **text modality**, we utilize the `nlpaug` (Ma, 2019) library, to add different types of noise. The user has option to choose a subset of noise types from: 1) Keyboard noise that simulates keyboard distance error; 2) OCR noise that simulates OCR engine noise; 3) Random character noise to insert, substitute, or delete random characters; 4) Antonym noise to swap random words with their antonyms; 5) Random word noise to insert, substitute, or delete random words; 6) Spelling noise to add spelling mistakes according to spelling mistake dictionary; and 7) Back-translation noise to translate the text to another language, and then translate back to English. The parameters of these noise types can be specified by the user, and are transferred to the `nlpaug` library for adding the specific type of noise.

---

[10]https://github.com/iver56/audiomentations

For **image modality**, we added different types of noise suggested and implemented by Hendrycks & Dietterich (2018). 15 perturbations are included such as: adding Gaussian noise, shot noise, impulse noise, defocus blur, frosted glass blur, motion blur, zoom blur, snow, frost, fog, changing the brightness, contrast, elasticity, pixelating, and JPEG compressing.

**Label Noise**: Another way to insert aleatoric uncertainty is to add label noise in the data (i.e., randomly switch the labels of some samples). Since the uncertainty induced by this type of noise cannot be reduced with additional amount of data, this noise shall increase the aleatoric uncertainty of the model. To insert this type of noise, we choose samples at random (according to a probability defined by the user), then, based on their deep features (Wav2Vec (Baevski et al., 2020) for audio, BERT (Devlin et al., 2019b) for text and VGG-11 (Simonyan & Zisserman, 2015) for images), we find the average distance to the 5 nearest points from each class, and assign the label of the class that has the closest mean distance.

**OOD Injection**: Ideally, the models shall be uncertain on data points from unknown distribution (i.e., distribution they haven't been trained on). In the literature, often the OOD samples are taken from another dataset, which can simplify the problem, because such samples are far from the training data. For this matter, we kept a separate set of samples from the same dataset, but belonging to classes that are not present in the training data, as OOD samples.

# 4 BASELINE MODELS WITH UNCERTAINTY QUANTIFICATION

## 4.1 BASELINE MODELS

We develop baseline models with three different uncertainty quantification algorithms, to serve as a starting point for other research and benchmarking initiatives. For the sake of simplicity, we choose late or decision fusion approaches, where we have classification networks for each modality, and then fuse their decision by simple averaging the output logits. These baselines were selected to instantiate unimodal and multimodal architectures, which can be trained on the dataset and are not intended to serve as a comprehensive benchmark, nor did we endeavor to achieve the best possible performance.

For the image modality, we used a simple convolutional neural network, as depicted in Figure 7. For the audio modality, we extracted 128x128 mel-spectrograms from padded audio samples, and used a convolutional network for classification, as depicted in Figure 1. For the text modality, we extracted the BERT (Devlin et al., 2019a) embeddings for each token, and averaged them out, so that we have one embedding per text passage. Then, we passed the embedding through a simple feed-forward neural network (Figure 7) to get the predictions. As depicted in Figure 7, each model includes two output heads: one for the prediction and the other for aleatoric uncertainty, following the methodology outlined by Valdenegro-Toro & Mori (2022). Then, to combine the aforementioned unimodal networks into a multimodal architecture, we adopted the late fusion approach. In the Monte Carlo Dropout and Deep Ensemble methods, we obtained the multimodal prediction by averaging the logits from the final layers of the classifiers. For the RCML, we modified the output of the last layer in each network to produce evidence, as described in (Xu et al., 2024), and followed their methodology for combining the evidence.

The dropout probability is 0.3, with the deep ensemble comprising 10 networks. Networks were trained for up to 300 epochs, with early stopping after 10 epochs of no validation loss improvement.

## 4.2 UNCERTAINTY METRICS

For uncertainty quantification, we implemented 3 approaches: Monte Carlo Dropout (MCD) (Gal & Ghahramani, 2016), Deep Ensemble (DE) (Lakshminarayanan et al., 2017), Reliable Conflictive Multi-View Learning (RCML) (Xu et al., 2024). In Monte Carlo Dropout and Deep Ensembles, we use the aleatoric entropy and the epistemic entropy as as uncertainties measures $H_{\text{Ale}}(y \mid \mathbf{x}) = \text{entropy}(p_{\text{Ale}}(y \mid \mathbf{x}))$ and $H_{\text{Epi}}(y \mid \mathbf{x}) = \text{entropy}(p_{Epi}(y \mid \mathbf{x}))$, where $p_{Epi}$ and $p_{Ale}$ are the probabilities obtained according to (Valdenegro-Toro & Mori, 2022). In RCML, we measure the aleatoric uncertainty with the expected entropy such as:

$$\mathbb{E}_{p(\boldsymbol{\pi}|\mathbf{x},\hat{\boldsymbol{\theta}})}[H[P(y \mid \boldsymbol{\pi})]] = -\sum_{k=1}^{K} \frac{\alpha_k}{\alpha_0} \left(\psi\left(\alpha_k + 1\right) - \psi\left(\alpha_0 + 1\right)\right), \quad (2)$$

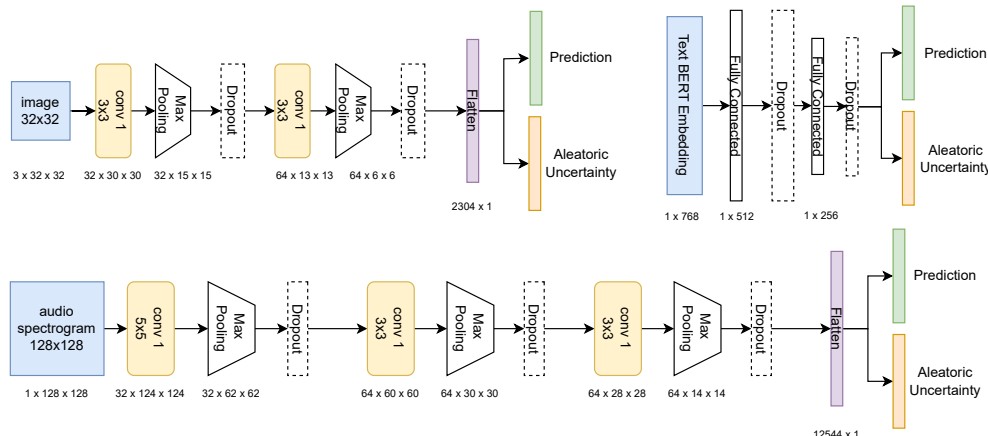

Figure 7: The classification network for the text modality.

where $\alpha_k$ is the $k$-th concentration parameter of the Dirichlet distribution, and $\alpha_0$ is the sum of all concentration parameters. $\psi$ is the digamma function. As a measure for epistemic uncertainty, we take $\frac{N}{\alpha_0}$, where $N$ is the number of classes. We evaluate the measures of accuracy and uncertainty of the models on the clean dataset, the dataset with reduced diversity ($\searrow$ Diversity), the dataset with increased sample noise ($\searrow$ Sample Noise), and the dataset with switched label $\nearrow$ Label Noise).

## 4.3 RESULTS

The results are summarized in Tables 1. (For noise generation parameters, please refer to the Appendix C.1). Since in RCML the uncertainty is quantified differently compared with MCD and DE, we cannot directly compare their values, and relative changes to the two type of uncertainty compared to the clean dataset is reported in the table. We observe that accuracy always decreases with increasing the label noise, but reducing diversity and increasing sample noise may not always decrease accuracy in the image modality.

Table 1: Results for UQ with baseline models. The absolute values are reported for clean dataset, and changes in percentages relative to clean dataset are reported for the noisy versions of LUMA dataset.

| Method | Clean | | $\searrow$ Diversity | | $\nearrow$ Label Noise | | $\nearrow$ Sample Noise | |
|---|---|---|---|---|---|---|---|---|
| | Ale. | Epi. | Ale. | Epi. | Ale. | Epi. | Ale. | Epi. |
| MCD Image | 1.00 | 1.03 | -15.73% | -11.66% | +59.20% | +54.51% | +4.44% | +2.18% |
| MCD Audio | 0.52 | 0.70 | -5.54% | +2.16% | +96.63% | +54.49% | +23.12% | +14.40% |
| MCD Text | 0.37 | 1.01 | -3.91% | -2.62% | +93.59% | +2.41% | +64.96% | -2.03% |
| MCD Multi. | 0.26 | 0.78 | -8.52% | -1.21% | +122.44% | +11.60% | +59.14% | +9.89% |
| DE Image | 1.45 | 1.40 | -37.49% | -8.54% | -7.43% | +0.24% | -18.46% | -3.22% |
| DE Audio | 0.56 | 0.99 | -27.39% | -3.34% | **+156.40%** | +50.43% | **+70.26%** | +34.41% |
| DE Text | 0.42 | 1.01 | +5.02% | -6.15% | +81.26% | -0.51% | +62.24% | -7.11% |
| DE Multi. | 0.31 | 0.82 | -22.80% | -3.40% | +115.15% | +20.62% | +45.97% | +5.54% |
| RCML Multi. | 1.99 | 0.43 | **+8.34%** | **+16.16%** | +64.72% | **+106.16%** | +36.19% | **+58.21%** |

Table 2: OOD Detection AUC Values for Different Methods

| Method | MCD | | | | DE | | | | RCML Multi. |
|---|---|---|---|---|---|---|---|---|---|
| | Image | Audio | Text | Multi. | Image | Audio | Text | Multi. | |
| AUC | 0.54 | 0.47 | 0.53 | 0.50 | 0.54 | 0.49 | 0.54 | 0.50 | **0.91** |

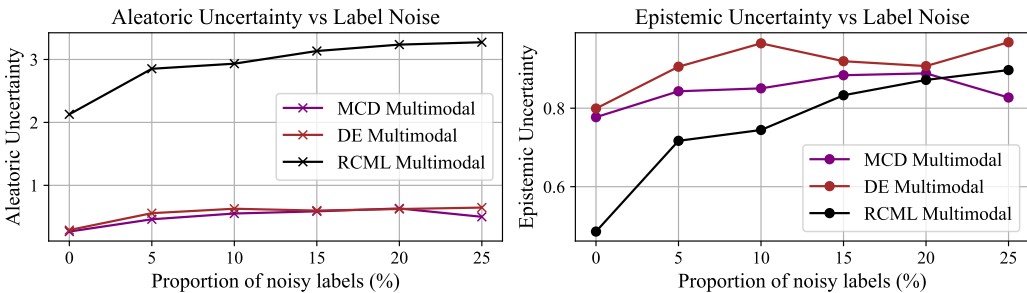

Figure 8: Changes in uncertainty estimations under different proportions of label noise (in percentages). The RCML (Reliable Conflictive Multi-View Averaging) approach consistently increases both aleatoric and epistemic uncertainties with increased label noise. In contrast, the MCD (Monte Carlo Dropout) and DE (Deep Ensemble) models sometimes fail to increase the corresponding uncertainty estimations in this experiment.

As we can observe in the table, in most cases, adding label and sample noises effectively increases the epistemic and aleatoric uncertainties. Interestingly, in most MCD and DE models, the uncertainty decreases when they are trained on data with lower diversity. This may indicate that these approaches fail to recognize data points outside their training distribution, which we will further investigate with the OOD detection task.

We evaluate AUC score for OOD detection by the networks based on the epistemic uncertainty. The results are summarized in Table 2. We can see that Monte Carlo Dropout and Deep Ensembles fail to provide epistemic uncertainty values suitable for OOD detection in LUMA dataset, with a poor performance of approximately 0.5 AUC value. On the other hand, the RCML achieves an outstanding AUC score of 0.91, indicating that the epistemic uncertainty values quantified with this method can be effectively used for OOD detection.

To further evaluate the qualities of the uncertainties of the different models, we estimate the epistemic and aleatoric uncertainties under different amounts of label noise. Ideally, we expect a good uncertainty quantification algorithm to provide higher uncertainty values for more noisy data. As we can see from Figure 8, only RCML consitently raises the uncertainty estimates under increased label noise, which again shows the higher quality of its uncertainty estimates over the other baselines.

In conclusion, the performance of Monte Carlo Dropout and Deep Ensembles indicates a limitation in their suitability for OOD detection in LUMA dataset. This suggests new avenues for further exploratory research to leverage uncertainty estimation for robust detection of out-of-distribution samples. Furthermore, the observed disparities highlight the necessity for a comprehensive benchmarking effort on LUMA dataset, encompassing a broader array of state-of-the-art methodologies.

## 5 CONCLUSION

In this paper, we propose LUMA, a new multimodal dataset for learning from uncertain and multimodal data and benchmarking. It includes image, audio and text modalities and a Python package for compiling different versions of the dataset with various amounts and types of uncertainty and noise.

The dataset can be easily extended with additional modalities and augmented with more data samples. The open-source nature of the data compilation pipeline and code for uncertainty and noise generation facilitates the integration of new contributions from the community to promote multimodal uncertainty studies and benchmarking initiatives.

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

APPENDICES

## A    DATASET LINKS AND LICENSES

In this section we present the dataset links and licenses for the LUMA dataset.

- LUMA Dataset link[11]:
  `https://osf.io/8ph6y/?view_only=8272969d8cd34c0b9459659fb4f41507`
- LUMA Dataset DOI: Not included to keep double blind nature of the peer-review. Will be added in the camera-ready version in case of acceptance.
- LUMA Python package and baseline code:
  `https://osf.io/8ph6y/?view_only=8272969d8cd34c0b9459659fb4f41507`
- LUMA Dataset license: `CC BY-SA 4.0`
- LUMA Code license: `GPL-3.0`

## B    DETAILS ABOUT DATA COLLECTION

### B.1    IMAGE MODALITY

#### B.1.1    LICENSES OF THE SOURCE DATASETS

As mentioned in Section 3.2 of the main paper, we use the CIFAR-100 and CIFAR-10 (Krizhevsky, 2009) datasets and EDM (Karras et al., 2022) generated images from the same distribution for the image modality.

Unfortunately, we failed to find an explicit license for the CIFAR-10/100 datasets, hence, we did not include the images in our dataset repository, and instead, allow the users to download them directly using the provided LUMA dataset compilation and uncertainty generation tool.

The EDM Generated images retrieved from https://huggingface.co/datasets/P2333/DM-Improves-AT are published under the `Apache-2.0` license.

### B.2    AUDIO MODALITY

In this section, we will present some additional information regarding data collection process of audio modality.

#### B.2.1    LICENSES OF THE SOURCE DATASETS

As mentioned in Section 3.2 of the main paper, we collected audio data from 3 sources: The Spoken Wikipedia (Köhn et al., 2016), LibriSpeech (Panayotov et al., 2015), and Mozilla Common Voice (Ardila et al., 2020) corpora.

The Spoken Wikipedia dataset is published under the `CC BY-SA 4.0` license. Following the requirements of this license, we also distribute our dataset under the same license. The audio samples extracted from this dataset are located in the `sw_audio` directory in LUMA dataset.

The LibriSpeech dataset is published under the `CC BY 4.0` license. The audio samples extracted from this dataset are located in the `ls_audio` directory in LUMA dataset.

The Mozilla Common Voice dataset is published under the `CC0` (public domain) license. The audio samples extracted from this dataset are located in the `cv_audio` directory in LUMA dataset.

#### B.2.2    AUDIO EXTRACTION AND VALIDATION

To extract the audio pronunciation of each word as described in Section 3.2, we follow the Algorithm 1. First, the algorithm iterates through all the words in the transcripts. If a word matches one of the

---

[11]The dataset and code links are anonymized for the peer-review process. They will be substituted with public links in case of acceptance.

labels or their plural forms, it is added to the `extraction_words` list (Lines 3-10), along with its corresponding time steps and audio path. Next, the algorithm processes each word in this list, using the `extract_audio` function to extract the relevant audio segments. These segments are then added to the `extracted_audios` list, which is returned as output of the algorithm (Lines 11-17).

---

**Algorithm 1** Audio Extraction Algorithm

---

1: **Input:** labels ← list of the CIFAR-100/10 labels
2: **Input:** transcripts_paths ← list of (transcript, audio_path) tuples
3: extraction_words ← empty list
4: **for** transcript, audio_path in transcripts_paths **do**
5:     **for** word in transcript **do**
6:       **if** word in labels **or** word + 's' in labels **or** word + 'es' in labels **then**
7:         extraction_words.append((word, transcript[word].start_time, transcript[word].end_time, audio_path))
8:       **end if**
9:     **end for**
10: **end for**
11: extracted_audios ← list
12: **for** entry in extraction_words **do**
13:     word, start_time, end_time, path ← entry
14:     audio_segment ← extract_audio(path, start_time, end_time)
15:     extracted_audios.append(audio_segment)
16: **end for**
17: **return** extracted_audios

---

Then, we perform automatic validation with the OpenAI's Whisper Large V3 model (Radford et al., 2023), and on the subset of the negative predictions of the model we perform a manual validation. For the manual validation, we have collected manual annotations thanks to the efforts of our volunteering colleagues and friends. To build the annotation interface, we opted for the Label Studio (Tkachenko et al., 2020-2022), an open-source data labeling platform. You can see the screenshot of the annotation interface in Figure 9. In total, we collected 16,744 manual annotations from 17 human annotators over a period of 2 months. Each sample was annotated by 2 annotators, and only annotations with 100% agreement were accepted.

After automatic and manual validation, we have 135,096 audio samples with the class distribution shown in Figure 10.

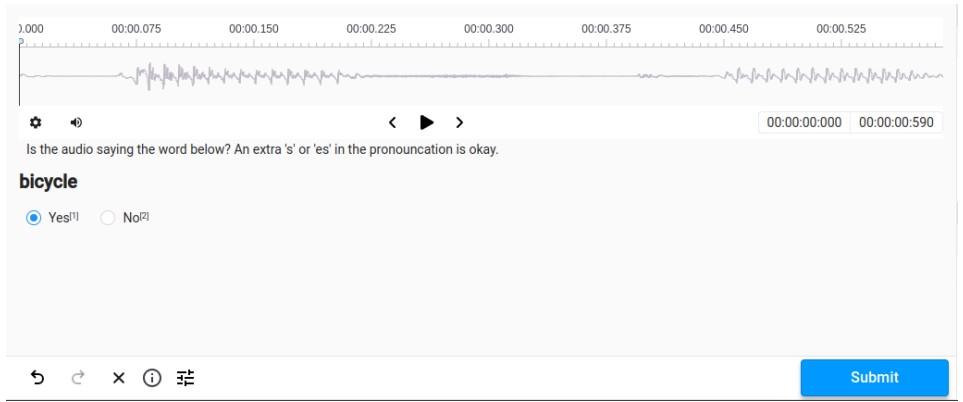

Figure 9: The labeling interface using Label Studio

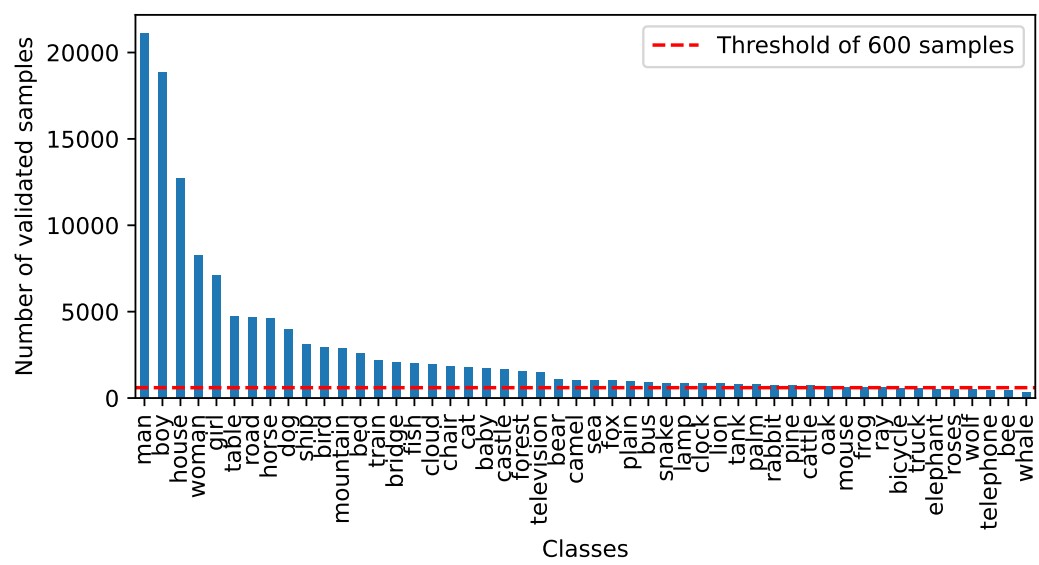

Figure 10: Number of validated audio samples per class. We will include the classes with higher than 600 samples as in-distribution data, and others as out-of-distribution data.

### B.3 TEXT MODALITY

For the text modality, as described in the Section 3.3, we employed Google's Gemma-7B Instruct model (Mesnard et al., 2024) to generate more than 1,200 text samples per class, using 13 different prompts. The 13 prompts are as follows:

```
"You are talking with your friend about some topic. Use the word <
    word> in a sentence with your friend. Use casual language.
    Tone: Casual / Conversational, length: short",
"You are the prime minister of the United Kingdom. During a press
    conference you are asked a question about <word>. Give a
    sentence from that press conference mentioning the word <word>.
     Tone: Formal, length: medium",
"You are explaining a five year old child what the word <word>
    means. Use very simple and explanatory language, so the kid
    will understand the meaning of the word <word>. Tone: Casual,
    complexity: simple",
"Imagine you are writing a science fiction book. Write a
    conversation from that book mentioning the <word>.",
"You are the editor in a mainstream journal. Write a sentence from
     a news article about a <word> in your journal that mentions
    the word <word>.",
"You are a teenager writing a post in Facebook about <word>. Write
     the post about the experience you had with the <word>.",
"You are playing a word describing game with your friend. The word
     is <word>, and you shall describe it without mentioning the
    word itself, so your friend will guess it. Explain it to him
    clearly in a simple language.",
"Think of something else that shares similar characteristics or
    functions with the <word>. Draw comparisons or use analogies
    between that other word and the <word>.",
"Place the word <word> within historical context. How would you
    describe it in relation to its origins, evolution, or
    significant historical events? Be creative in your description
    .",
```

```
"Consider how the word '<word>' is depicted or referenced in
    popular culture, literature, or media. Describe it by
    referencing these cultural elements.",
"Pretend you are a character who sees a <word> for the first time
    in your life. Describe it from the character's perspective,
    considering their background, personality, and knowledge.",
"Write a 4 line small poem about the word <word>. Be creative, and
    use casual tone for the poem.",
"You are a musician composing a song inspired by <word>. Write the
    lyrics to the song, capturing the mood, emotions, and imagery
    associated with <word>. Use rhythm and melody to convey the
    essence of <word> in your music."
```

The `<word>` was replaced with the class labels. After generating the text data, to check if the generated text has any bias or stereotype, we again used the Gemma-7B Instruct model with the following prompt:

```
Your job is to identify biases in texts. You will be given a text,
    and you need to classify it into one of: [gender_bias,
    cultural_bias, racial_bias, religous_bias, no_bias].
Give the output in JSON format: {{"bias_type": <predicted bias tpe
    >}}. DO NOT WRITE ANYTHING ELSE. \n\n The text is: <text>
```

The `<text>` was replaced with the text that needed to be checked. The number of biased and unbiased texts in each class according to Gemma model can be found in Figure 11. As we can see, the labels man, woman, boy, and girl have high amount of biased texts. Some examples of identified biased texts are:

```
A woman is a grown-up person who has a soft, nurturing personality
    . She usually takes care of her family and friends, and
    sometimes works outside the home. Women are strong and smart,
    they can do many things that men can do.

Man, a force of might,
A guardian, protector, and light.
With strength and wisdom, they stand tall,
Strong and proud, answering the call.

A tool in a toolbox is an efficient and valuable asset that aids
    in various tasks. Similarly, a man is also a valuable asset to
     any group or society. Just like a tool in a toolbox, a man's
    capabilities are tailored to fulfill different roles and
    functions, making him an essential component of any endeavor.
```

To alleviate this issue, for the labels of man, woman, boy and girl, we reconstructed the prompts for text generation. The prompt used is the following:

```
Write two sentences with topic: <topic>, and keywords: <keyword1>,
    <keyword2>.
```

where `<topic>` was replaced with one of the following words: 'factual', 'fiction', 'history', 'books', 'movies', 'philosophical', and `<keyword1>` with one of the following words: 'man', 'woman', 'boy', 'girl'. `<keyword2>` was treated differently depending on the label: For the term 'man', it was replaced with one of the following words:

```
actor, king, scientist, doctor, wizard, duke, lord, governor,
    prime minister, father, sorcerer, waiter, chess, director,
    producer, uncle, singer
```

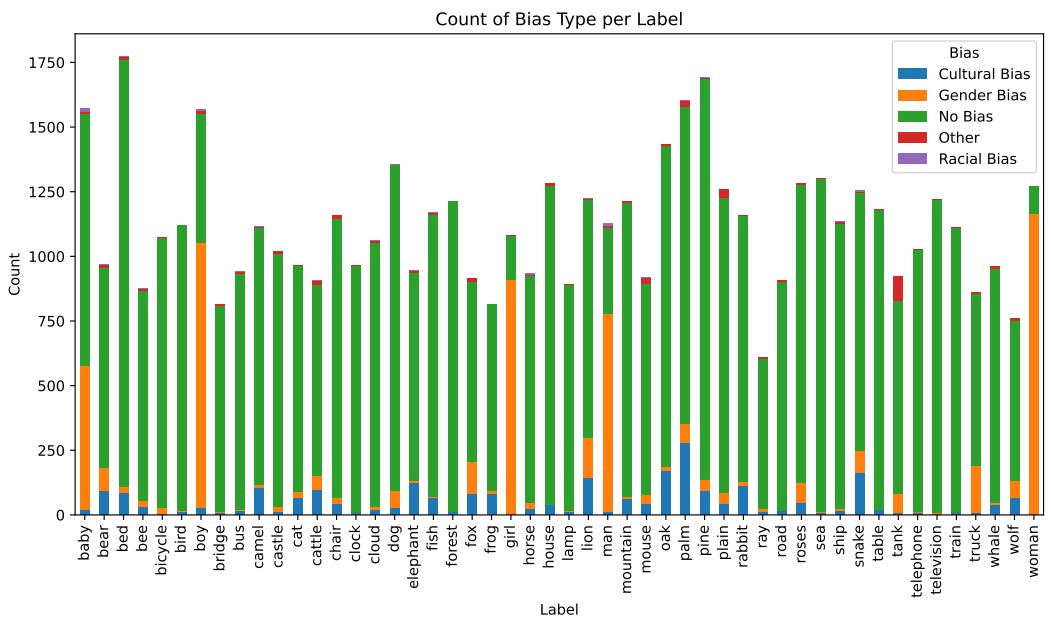

Figure 11: The amount of texts with different biases according to Gemma-7B Instruct model.

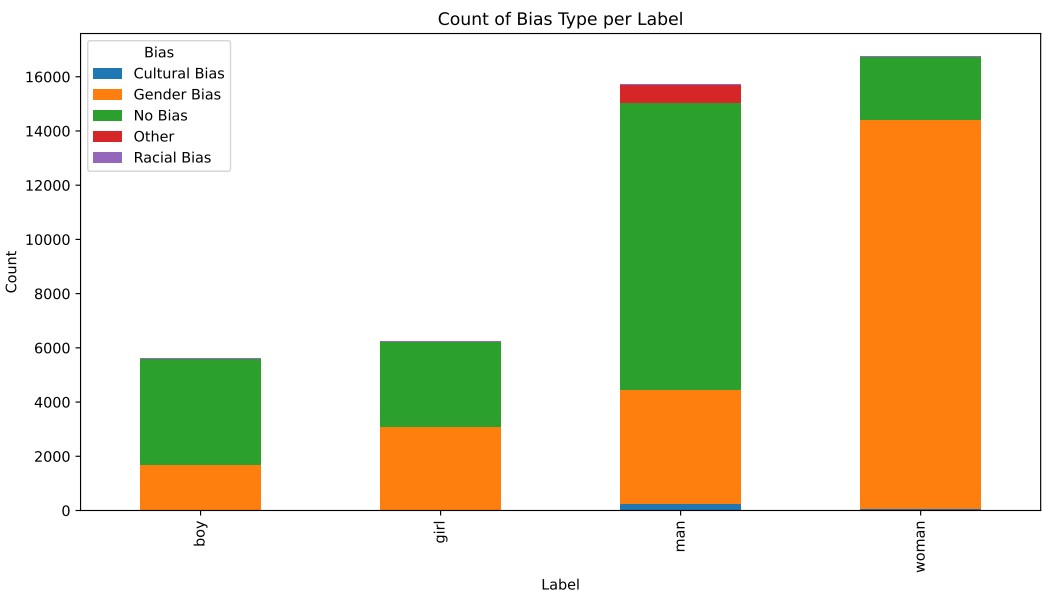

Figure 12: The amount of texts with different biases according to Gemma-7B Instruct model, after reconstructing the prompts for these 4 labels. Although there is still high amount of bias, we can filter them out and still have enough unbiased texts (more than 800 text passages per class).

For the term `woman`, it was replaced with one of the following words:

```
actress, queen, scientist, doctor, witch, duchess, lady, governor,
    prime minister, mother, sorcerer, waitress, chess, director,
    producer, aunt, singer
```

For the term `boy`, it was replaced with one of the following words:

```
kid, actor, prince, son, nephew, pupil, student, singer
```

And for the term `girl`, it was replaced with one of the following words:

```
kid, actress, princess, daughter, niece, pupil, student, singer
```

We then performed another round of bias detection using the Gemma model. While we found that a significant amount of bias still exists, we identified enough unbiased texts (according to the Gemma model) to include in the LUMA dataset. The number of biased and unbiased texts after re-generating the data for these 4 classes can be found in Figure 12.

To validate that the generated text passages could be correctly classified, we fed them back into the Gemma model using the following prompt:

```
In the text: "<text>", you need to understand what is the <
    masked_word> or find out what is the whole text about.
You need to classify the text into one of these classes: "<classes
    >"
give the output in JSON format: {{"class": <predicted class (1
    word from the list)>}}. DO NOT WRITE ANYTHING ELSE.
```

were the `<text>` was replaced with the text that needed to be verified, and the `<classes>` was replaced with the possible classes.

## C  DETAILS ABOUT BASELINE ARCHITECTURES AND DATASETS

| Parameter | Clean Dataset | Reduced Diversity | Increased Label Noise | Increased Sample Noise |
|---|---|---|---|---|
| **Compactness** | 0 | 20 | 0 | 0 |
| **Sample Noise** | False | False | False | True |
| **Label Switch Prob.** | 0 | 0 | 0.3 | 0 |
| **Noisy Data Ratio** | 0.0 | 0.0 | 0.0 | 1 |
| **Audio Noise SNR** | None | None | None | 3-5 |
| **Image Noise** | None | None | None | Gaussian Noise: 4, Shot Noise: 4, Impulse Noise: 4, Defocus Blur: 4, Frosted Glass Blur: 4, ... (additional noises summarized) |
| **Text Noise** | None | None | None | KeyboardNoise: 1-5 char, 3-8 word, BackTranslation, Spelling: 0.4, OCR: 0.5 word |

Table 3: The parameters for generating the different versions of the dataset for experiments with the baseline models.

## C.1 DATASET GENERATION

We compiled one clean version and three noisy versions of the dataset, each with a specific type of noise: reduced diversity, increased label noise, and increased sample noise. The parameters for compiling the different versions can be found in Table 3. These parameters were selected to introduce sufficient noise, ensuring observable changes in the uncertainty metrics. Please refer to our codebase for the full configuration files.

## C.2 CLASSIFICATION ACCURACIES FOR BASELINE MODELS

In Table 4 we present the classification accuracy measures for the clean dataset and variations in accuracy under different types of noise for the three baseline models (Monte Carlo Dropout, Deep Ensemble, and RCML) across each modality. Each version of the noisy dataset incorporates a single type of noise: Reduced Diversity (K=20), Label Noise (30%), and Sample Noise, with severity varying by noise type and modality (detailed in Appendix C.1). The RCML model, a multimodal approach, is evaluated exclusively in the multimodal setting. We observe that accuracy always decreases with increasing the label noise, but reducing diversity and increasing sample noise may not always decrease accuracy in the image modality.

Table 4: Classification accuracies for the clean dataset and variations in accuracy under different types of noise.

| Model | Clean Dataset | Reduced Diversity | Increased Label Noise | Increased Sample Noise |
|---|---|---|---|---|
| | Accuracy | Difference in accuracy from the clean dataset | | |
| MCD Image | 0.335 | +0.058 | -0.306 | +0.019 |
| MCD Audio | 0.867 | -0.025 | -0.784 | -0.155 |
| MCD Text | 0.965 | -0.027 | -0.864 | -0.144 |
| MCD Multi. | 0.991 | -0.010 | -0.874 | -0.063 |
| DE Image | 0.387 | +0.066 | -0.166 | +0.019 |
| DE Audio | 0.912 | -0.003 | -0.809 | -0.149 |
| DE Text | 0.973 | -0.023 | -0.864 | -0.125 |
| DE Multi. | 0.996 | -0.006 | -0.849 | -0.042 |
| RCML Multi | 0.973 | -0.128 | -0.833 | -0.148 |

