# OpenReview forum: "LUMA: A Benchmark Dataset for Learning from Uncertain and Multimodal Data"
_ICLR.cc/2025/Conference — ICLR 2025 Conference Withdrawn Submission_

### Official Review · Reviewer_khBJ · 2024-10-31

**Soundness:** 2
**Presentation:** 2
**Contribution:** 2
**Rating:** 3
**Confidence:** 3

**Summary:**

This paper attempts to build a benchmark dataset for Multimodal Uncertainty Quantification (MUQ). It creates the LUMA dataset which contains multimodal data. At the same time, it builds baseline models on this new dataset.

**Strengths:**

Overall, the paper is clearly written. However, there are some limitations.

**Weaknesses:**

1. This paper lacks practical applications in real-world scenarios. Both the data and the noise are in a simulated environment, which is very different from real-world applications. At the same time, it collects images from CIFAR, audio using keyword spotting techniques, and text using LLM to generate short descriptions. However, I cannot imagine that there is multimodal data containing these types of data in real-world scenarios.
2. All sample noise and label noise are simulated, but no real noisy data is used. Therefore, this dataset is far from real-world scenarios.
3. Section 4 only covers three baselines: MCD, DE, and RCML. Please cover more approaches.
4. This paper contains some minor grammatical errors. Please proofread this article:
a)	described in the section 3.4.1, => Section
b)	73 Additional recordings of => 73 additional
c)	Please abbreviate words only once.

**Questions:**

Please refer to the weakness part.

---

> ### Author Response · Authors · 2024-11-19
>
> Dear Reviewer, thank you for the thorough review of the paper. We would like to address some of your concerns below:
>
> 1. We would like to emphasise that the LUMA dataset is primarily designed as an experimental tool for researchers working on the development of new multimodal uncertainty quantification algorithms. The purpose of the dataset is to provide a 3-modal dataset along with tools to inject real-world-like noise within a controlled environment. This setup allows researchers to study uncertainty quantification methods without the complexities that arise in real-world data collection. By offering a controlled, simulated setting, LUMA enables researchers to focus on methodological advancements that could later be applied and tested in real-world contexts.
>
> 2. In designing the dataset, we aimed to simulate realistic noise conditions as closely as possible. For image data, we incorporated elements like motion blur and JPEG compression noise; for text, we included spelling errors and translation noise; and for audio, we added background conversations, typing sounds, laughter, and more. We opted for a synthetic noise approach because it provides researchers with greater control over the types and levels of noise introduced, allowing them to systematically study and evaluate their uncertainty quantification algorithms. Using an inherently noisy dataset would reduce this level of control, thereby limiting the dataset’s utility for detailed, iterative experimentation.
>
> 3. We would like to emphasize that the primary goal of this paper is to introduce the LUMA dataset and the noise injection tool rather than to serve as an extensive benchmarking study. The baseline models included—MCD, DE, and RCML—are intended to offer a foundational reference and starting point for researchers who wish to experiment with the LUMA dataset. By including these baselines, we aim to demonstrate the dataset's utility and provide a preliminary framework. However, we encourage future studies to explore other multimodal approaches, such as multimodal Gaussian [1] and neural processes [2], or other evidential models [3, 4]. Additionally, we hope that the LUMA dataset will inspire the design of new studies and methodologies specifically tailored to multimodal uncertainty quantification, further advancing this area of research.
> 4. We thank the reviewer for pointing out the grammatical errors, we fixed the errors identified by the reviewer and will proofread the article once more to eliminate any remaining mistakes.
>
> [1] Jung, M. C., Zhao, H., Dipnall, J., Gabbe, B., and Du, L. (2022). Uncertainty estimation for multi-view data: The power of seeing the whole picture. Advances in Neural Information Processing Systems, 35:6517–6530.
>
> [2] Jung, M. C., Zhao, H., Dipnall, J., and Du, L. (2024). Beyond unimodal: Generalising neural processes for multimodal uncertainty estimation. Advances in Neural Information Processing Systems, 36.
>
> [3] Huang, L., Ruan, S., Decazes, P., and Denœux, T. (2025). Deep evidential fusion with uncertainty quantification and reliability learning for multimodal medical image segmentation. Information Fusion, 113:102648.
>
> [4] Liu, W., Yue, X., Chen, Y., and Denoeux, T. (2022). Trusted Multi-View Deep Learning with Opinion Aggregation. Proceedings of the AAAI Conference on Artificial Intelligence, 36(7):7585–7593.

---

> > ### Comment · Reviewer_khBJ · 2024-11-26
> > **Reply to Authors**
> >
> > Thanks very much for your response. I understand the motivation behind this dataset, i.e., providing a resource for multimodal uncertainty learning. However, this synthetic dataset is far from real-world environments. It is hard to imagine a scenario where the multimodal data pairs (image, audio, text) provided in this dataset would be found in the real world. In my view, AI should serve as a tool to solve real-world problems. Unfortunately, the paper does not address this issue: whether models that perform well on this dataset can also be effectively applied to real-world tasks. Therefore, I decide to maintain my score.

---

### Official Review · Reviewer_djXn · 2024-11-03

**Soundness:** 2
**Presentation:** 2
**Contribution:** 2
**Rating:** 5
**Confidence:** 4

**Summary:**

The paper presents LUMA, a multimodal dataset designed explicitly for benchmarking uncertainty in multimodal learning algorithms featuring audio, image, and text data. The author sources images from CIFAR 10/100, audio samples (consisting of label information) from diverse labeled speech corpora such as LibriSpeech, Spoken Wikipedia, and Common Voice, and text generated synthetically via Gemma-7B Instruct LLM. The authors also provide a Python package to customize uncertainty levels through controlled noise generation across all the modalities. Additionally, the paper introduces baseline models incorporating Monte Carlo Dropout, Deep Ensemble, and Reliable Conflictive Multi-View Learning.

**Strengths:**

1. This paper introduces a novel benchmark, LUMA, designed to measure uncertainty across multiple modalities: audio, image, and text. Additionally, the LUMA dataset allows for controlled manipulation of different types and levels of uncertainty.

2. The authors provide baseline pre-trained models along with three uncertainty quantification methods—Monte Carlo Dropout, Deep Ensemble, and Reliable Conflictive Multi-View Learning—offering a solid starting point for benchmarking.

**Weaknesses:**

1. The choice of data sources for specific modalities is unclear. For instance, it’s not evident why CIFAR-10/100 was used for images rather than ImageNet, which includes 1,000 labels and could provide greater diversity for benchmarking. I suggest that the authors clarify this decision in the main paper.

2. To convert text labels into audio, the authors propose a complex approach involving mapping labels to utterance transcriptions and segmenting using forced alignment. However, a simpler alternative might be to synthesize text labels directly with state-of-the-art Text-To-Speech models. Did the authors compare their approach with synthetic audio generation as described in Section 3.2?

3. The paper is difficult to follow. For example, Section 4 covers quantitative results of various baselines on LUMA but combines details on uncertainty algorithms, results, and analysis without clear distinctions, making it challenging to understand. Additionally, critical information on model architecture and uncertainty algorithms seems to be either buried in the Appendix or missing. Clearer organization in the main text would improve readability.

4. The paper lacks error analysis and qualitative results. If these are included in the Appendix, I suggest mentioning them in the main paper for accessibility.

**Questions:**

Please refer to the weakness section.

---

> ### Author Response · Authors · 2024-11-19
>
> We thank the reviewer for the thorough review and comments. We would like to address some of the comments below:
>
> 1. The dataset is primarily designed as an experimental tool for researchers focusing on multimodal uncertainty quantification. Using ImageNet would significantly increase the model's training and inference times due to its larger scale, which would reduce the dataset's practicality for iterative experimentation. Additionally, the complexity of managing 1,000 labels can hinder efficient debugging and make it more challenging to trace and analyze uncertainties effectively. Our selection of CIFAR-10/100 is thus intentional, aiming to balance diversity with usability. CIFAR’s more manageable label count and smaller image sizes allow researchers to conduct faster experiments, refine their methods, and observe uncertainty behaviours with greater control and fewer computational constraints. This setup ultimately enables a more practical, hands-on approach to exploring multimodal uncertainty in research contexts, where rapid iteration and insight into model behaviour are essential.
>
> 2. Initially, we considered synthetic audio generation, as it would have streamlined the data collection process by bypassing the need for forced alignment and extensive manual validation. However, as far as we know, there is no model capable of delivering the diversity and quality needed for our dataset. For instance, while OpenAI’s Text-To-Speech API could produce data with up to six different voices, our dataset includes thousands of unique voices from individuals with varied backgrounds, intonations, and accents. This natural diversity in voice contributes significantly to the dataset's authenticity and enhances its suitability for studying real-world uncertainty in multimodal contexts.
>
> 3. Based on your recommendation we included more detailed information about the baseline architectures and re-structured the section in the main paper.
>
> 4. Since the paper is not intended to serve as a comprehensive benchmark study, we did not initially include an error analysis. However, we agree with the reviewer that incorporating error analysis can significantly enhance the usefulness and interpretability of the provided results. To address this, we have scheduled additional runs of the models to compute error bars. Once these runs are complete, we will include the error analysis either as a revision during the discussion period or, if it takes longer, in the camera-ready version of the paper, should it be accepted.

---

> > ### Comment · Reviewer_djXn · 2024-11-26
> >
> > Thanks for addressing my concerns. I have updated my score.

---

### Official Review · Reviewer_eRfy · 2024-11-07

**Soundness:** 3
**Presentation:** 3
**Contribution:** 2
**Rating:** 5
**Confidence:** 3

**Summary:**

This work introduces a multimodal benchmark, integrating paired data for image, audio, and text to support the development of more reliable multimodal approaches. The image data includes samples from both CIFAR and generated data from diffusion models. For audio data, it features recordings of individuals pronouncing the corresponding class labels. For text data, large language models (LLMs) generate explanations for the associated content. By utilizing and fusing information from all modalities, the benchmark enables predictive analysis while controlling for data diversity, modality-specific noise, and the inclusion of out-of-distribution samples. Finally, it provides results using a pre-trained model with three uncertainty quantification methods: Monte-Carlo Dropout, Deep Ensembles, and RCML. This work paves the way to build a more trustworthy and robust model.

**Strengths:**

(1) Careful Data Collection Design: The data collection process for image, audio, and text modalities leverages generative models to augment existing data while incorporating mechanisms to ensure the quality of the benchmark dataset. This aligned, multimodal dataset for classification serves as a valuable resource for a wide range of research applications.

(2) Open-Source Python Package: This work includes a robust, open-source Python package, enabling other researchers to easily build upon and extend its functionalities. Typically, it has functions to generate multiple variants of the dataset with controlling the diversity, the amount of noise for each modality, and adding out-of-distribution samples.

**Weaknesses:**

(1) Simplistic Task Design: The classification tasks in this benchmark are relatively simple and may not align with the complexities of recent multimodal research. Although the work emphasizes uncertainty analysis rather than state-of-the-art multimodal models, the current focus of the multimodal research community is on more challenging tasks that better represent real-world applications. As a result, this benchmark may have limited contributions to the fields of trustworthy and robust machine learning. For instance, contemporary audio and vision models likely achieve very high accuracy on the benchmark’s classification tasks.

(2) Limited Depth of Analysis: The benchmark relies on relatively small pre-trained models, whereas larger and more powerful models might yield different insights and conclusions. As a benchmark paper, a more comprehensive analysis of various baselines, noise-handling methods, and technical details for uncertainty estimation would strengthen the work. Currently, it primarily explores late fusion-based multimodal methods with fixed pre-trained models, which narrows the scope of its conclusions and limits potential findings in robust machine learning. There are still one page space that more analysis results should be filled in. More analysis like why only RCML actually consistently raises the estimates under increased label noise and why RCML is diffferent from DE ad MCD can be further discussed.

**Questions:**

Diffusion models and LLMs for image and text generation, were used during the benchmark’s data collection. However, Gemma-7B, a relatively weak model, was employed in this process. It would be insightful to understand the potential impact of using a more powerful model, such as GPT-4o, to conduct the generation and whether there is a huge difference in data quality. Additionally, further analysis of the generated data could enhance the benchmark’s robustness. Currently, only audio data has dedicated visualization results in the paper. It would be beneficial to see more visualization and analysis for the text and image modalities as well.

Moreover, more additional information and details about the evaluation metric, uncertainty measurement setting, collected data examples can be introduced in the main section of the paper to help people understand more easily.

---

> ### Author Response · Authors · 2024-11-19
>
> We want to thank the reviewer for a thorough review of the paper and for identifying the strengths, areas of improvement and raising important points. Below we would like to discuss the points raised by the reviewer:
>
> 1. Regarding the task design, we would like to emphasize that the initial classification task is intentionally kept simple. This approach allows models to achieve high classification accuracy and low uncertainty, creating a clear baseline from which researchers can incrementally introduce complexity and observe how uncertainties evolve with various real-world-like noise injections. The purpose of the dataset is to study how uncertainties shift in response to different noise types. If the dataset were highly uncertain from the beginning, it would limit its flexibility to serve diverse experimental use cases and reduce researchers’ ability to tailor the noise levels to their specific studies.
>
> 2. We would like to clarify that the primary aim of this paper is to introduce the LUMA dataset, along with the associated noise injection tool. The models and benchmarks provided are intended as an initial reference point for researchers who wish to utilize the dataset, rather than as a comprehensive benchmark study. Our goal is to establish a foundation that researchers can build upon, exploring additional approaches as they leverage the dataset for multimodal uncertainty quantification studies.
>
> 3. Regarding our choice of Gemma-7B: At the time of dataset creation (Feb-April 2024), ChatGPT 4o had not yet been released. We conducted preliminary experiments with ChatGPT-4 but found that, for our specific task, the quality improvements in text generation did not justify the significantly higher costs associated with generating and validating the entire corpus using ChatGPT-4. Gemma-7B provided text quality that was more than adequate for our dataset's purposes, achieving the highest classification accuracy among the three modalities. This made Gemma-7B a more practical choice without compromising dataset utility.
>
> 4. Following the suggestion of the reviewer, we are also including the feature visualizations for the other two modalities, as shown in the Figure 6 of the revised paper.

---

> ### Comment · Reviewer_eRfy · 2024-11-24
>
> Thanks for your clarification and visualization. The additional figure looks great and I agree that the main motivation is to introduce the dataset itself. I have changed my score.

---

### Official Review · Reviewer_wJgm · 2024-11-07

**Soundness:** 3
**Presentation:** 3
**Contribution:** 2
**Rating:** 3
**Confidence:** 3

**Summary:**

This paper uses CIFAR image data, generates text and spoken word labels for these using a well described process, and creates a multimodal data set. It then describes software controlled processes and software to add noise to each of the three modalities. The datasets for the three modalities, noise tools, and benchmark code is made available.

**Strengths:**

Well known datasets from vision, text and audio are combined to create a multimodal dataset which contains the image, text and spoken word description of the image. There is a systematic process to add noise to each modality. Code is made available to add this noise.  The dataset for each modality is evaluated separately using baseline classifier models.Gender, Cultural and Racial Bias in labels are carefully avoided.

**Weaknesses:**

It is totally unclear how much use this dataset for a practical multimodal learning task. Also, this level of image with short caption or spoken word descriptions are largely bypassed by existing multimodal models.  Having spoken word labels (with varying amounts of noise) to low quality CIFAR images (with varying amounts of noise) does not appear to be very useful.

**Questions:**

The dataset is one-way in the sense that the images get you text and speech labels; and then noise is added. I cannot imagine how this would be used beyond relatively small academic exercises.

---

> ### Author Response · Authors · 2024-11-19
>
> Dear reviewer, thank you for your time and effort in reviewing our article, as well as for highlighting its strengths and raising insightful questions for further discussion. We would like to emphasize that this dataset was created as part of our research into multimodal uncertainty quantification. In our exploration, we identified a gap in available datasets that allowed researchers to explore different real-world-like noise scenarios and observe the resulting uncertainties in their algorithms. Consequently, this dataset is mainly dedicated to academic exploration and controlled experimentation, rather than for direct deployment in production models.
>
> We believe that this dataset can provide us with additional insights for advancing our understanding and interpretability of multimodal uncertainty quantification algorithms. Although the dataset itself is not designed for production use, we are confident that the algorithms and insights developed through this research will contribute meaningfully to real-world, safety-critical applications.

---

> > ### Comment · Reviewer_wJgm · 2024-11-26
> >
> > Dear authors, thank you for your response. I am very supportive of theoretical work focused at improving understanding, rather that creating tests on huge data. I also appreciate the desire to create a tool for injecting noise into clean data; but LUMA is still very restricted even for this limited purpose. If this was to be the main purpose of the paper, it should have been better motivated. I will maintain my review.

---

### Note · Authors · 2024-11-28

I have read and agree with the venue's withdrawal policy on behalf of myself and my co-authors.